# Metallization of Organically Modified Ceramics for Microfluidic Electrochemical Assays

**DOI:** 10.3390/mi10090605

**Published:** 2019-09-12

**Authors:** Ashkan Bonabi, Sari Tähkä, Elisa Ollikainen, Ville Jokinen, Tiina Sikanen

**Affiliations:** 1Drug Research Program, Faculty of Pharmacy, University of Helsinki, P.O. Box 56, 00790 Helsinki, Finland; ashkan.bonabi@helsinki.fi (A.B.); sari.tahka@helsinki.fi (S.T.); elisa.ollikainen@helsinki.fi (E.O.); 2Department of Chemistry and Materials Science, School of Chemical Engineering, Aalto University, Micronova, Tietotie 3, 02150 Espoo, Finland; ville.p.jokinen@aalto.fi

**Keywords:** organically modified ceramics, polymer metallization, adhesive bonding, electrochemical detection

## Abstract

Organically modified ceramic polymers (ORMOCERs) have attracted substantial interest in biomicrofluidic applications owing to their inherent biocompatibility and high optical transparency even in the near-ultraviolet (UV) range. However, the processes for metallization of ORMOCERs as well as for sealing of metallized surfaces have not been fully developed. In this study, we developed metallization processes for a commercial ORMOCER formulation, Ormocomp, covering several commonly used metals, including aluminum, silver, gold, and platinum. The obtained metallizations were systematically characterized with respect to adhesion (with and without adhesion layers), resistivity, and stability during use (in electrochemical assays). In addition to metal adhesion, the possibility for Ormocomp bonding over each metal as well as sufficient step coverage to guarantee conductivity over topographical features (e.g., over microchannel edges) was addressed with a view to the implementation of not only planar, but also three-dimensional on-chip sensing elements. The feasibility of the developed metallization for implementation of microfluidic electrochemical assays was demonstrated by fabricating an electrophoresis separation chip, compatible with a commercial bipotentiostat, and incorporating integrated working, reference, and auxiliary electrodes for amperometric detection of an electrochemically active pharmaceutical, acetaminophen.

## 1. Introduction

Owing to the low cost of the materials and the straightforward microfabrication protocols, photolithography of polymer-based photoresists is often considered the most feasible approach for fabrication of high-resolution microfluidic devices. Organically modified ceramics (ORMOCERs) are a new class of organic-inorganic hybrid polymers that can be patterned by a variety of techniques, including ultraviolet (UV) embossing, UV laser ablation, and UV lithography in proximity mode [1,2,3,4]. They comprise of organic side chains attached to an inorganic siloxane backbone [1,2], which yields excellent chemical and thermal stability [5] and inherent resistance to protein fouling [6,7]. The resulting microstructures are also optically transparent down to near-UV range [6] and the surfaces support cell adhesion in their native state [8]. Therefore, the use of ORMOCERs has attracted great interest in a range of optical [6,9,10] and biological and biomedical applications, including dental implants [11] and microfluidic sample handling [5,6,7] and cell culturing [8] devices. 

In UV lithography, the ORMOCERs act similar to standard negative-tone photoresists, such as SU-8. However, the UV exposure of ORMOCERs is typically conducted in proximity mode (instead of contact mode), which facilitates straightforward fabrication of rounded, concave cross-section profile upon controlled overexposure [9]. It has been demonstrated with the help of a commercial ORMOCER formulation, Ormocomp, that the microchannel cross-section shape can be flexibly tuned in a single lithographic step simply by adjusting the UV exposure dose, the distance of the proximity gap (between the photomask and the photoresist), and the layer thickness [9]. This gives appealing opportunities not only for implementation of concave micromirrors via deposition of thin-film aluminum on top of the rounded microstructures [9], but also for metallization of the microchannel side walls, which is generally very challenging for vertical-walled microstructures. High-quality step-coverage over microchannel edges typically requires rounding of corners and tilted side walls [12]. Although some work on integration of planar metallizations with ORMOCERs has been demonstrated in prior literature [13,14,15], fully developed processes feasible for integration of metals over three-dimensional topographies and facilitating subsequent sealing of the metallized microstructures are limited. Prior works report, for instance, evaporation of aluminum (Al) based micromirrors on concave ORMOCER microchannels [9] and pulsed laser deposition of silver (Ag) as an antibacterial coating on ORMOCER microneedles [16]. However, a large part of the prior literature on polymer metallization centers around the negative photoresist SU-8, which has well-established processes for, e.g., gold [17], palladium [18] and Al [19]. The adhesion strength, typically examined by pull-off tests, is largely affected by the surface chemistry, although surface roughness and topography also play a role. Often, incorporation of additional adhesion layers is required. For example, the adhesion strength between SU-8 and gold has been shown to increase by 75% through the use of titanium (Ti), OmniCoat or 4-aminothiolphenol as the adhesion layers [16]. Ti has also been used as the adhesion layer on ORMOCERs for deposition of copper based microwave circuits [15]. However, for custom polymers, even zero adhesion is sometimes reported necessitating either wet chemical treatment or plasma etching prior to metal deposition [20]. 

In this work, we focus on developing a set of metallization processes for Ormocomp. Key criteria for the processes are conductivity, adhesion, step coverage to allow implementation of electrodes over microchannels and a post-metallization bonding process that enables fabrication of enclosed microchannels with embedded metal elements. The prospected benefit of the developed protocols lies in the improved feasibility of ORMOCERs for implementation of standalone microfluidic and biomedical devices via integration of metal-sensing elements, such as thin-film electrodes or micromirrors. For the proof-of-concept experiments, a fully enclosed, Ormocomp-based electrophoresis separation chip featuring integrated working, reference, and auxiliary electrodes (made of Pt) for on-chip amperometric detection is fabricated and its performance in the intended purpose demonstrated with help of an electrochemically active pharmaceutical, acetaminophen.

## 2. Materials and Methods 

### 2.1. Fabrication of Ormocomp Microchannels 

The metallization processes were readily developed so as to ensure feasibility for step coverage over microchannel edges. The Ormocomp microchannels used as the substrates for metallizations were fabricated similar to previous work [9]. Briefly, two layers of Ormocomp (Microresist Technology Gmbh, Berlin, Germany) were sequentially spin coated on top of a glass substrate (Pyrex glass, thickness 350 µm, Plan Optik AG, Elsoff, Germany) and UV exposed on a MA-6 mask aligner (SÜSS MicroTec Inc, Garching, Germany). The first layer (15 µm thick) was spincoated at 6000 rpm for 30 s, flood exposed (76 mJ/cm²) and baked in the oven at 95 °C for 30 min (Figure 1A). Next, the second layer was spincoated at 2000 rpm for 30 s to yield a layer thickness of 35 µm, which defined the nominal microchannel height. This layer was exposed (19 mJ/cm²) in proximity mode using a gap distance of 400 µm, and baked in the oven at 95 °C for 30 min (Figure 1B). Both layers were developed simultaneously in OrmoDev developer (Microresist Technology Gmbh, Berlin, Germany) for 5 min to yield concave microchannels (Figure 1C), after which they were hard baked on a hotplate at 200 °C for 2 h. 

### 2.2. Metallization and Bonding of Ormocomp Microchannels

To integrate metal elements with Ormocomp microchannels, case-specific processes were developed for each of Pt, Au, silver (Ag), and aluminum (Al) by using Cr (for Al and Pt) or Ti (for Au and Ag) as the adhesion layers on the basis of prior literature [21,22,23,24]. The adhesion layer was first deposited on top of the microchannels by evaporation (Ti, 5 nm, IM-9912 evaporator, Instrumentti Mattila, Mynämäki, Finland) or sputtering (Cr, 17 nm, PlasmaLab 400, Oxford Instruments, Bristol, UK) (Figure 1D). Next, the metal layer was deposited also by evaporation (Al, Ag, Au) or sputtering (Pt) on top of the adhesion layer (Figure 1E) followed by vapour deposition of hexamethyldisilazane at 150 °C in the oven (Yield Engineering System, Livermore, CA, USA) for 30 min to promote adhesion of the photoresist to the metal layer. The detailed metallization parameters are listed in Table 1. The metal pellets for evaporation (Ti, Au, Ag, Al) were purchased from Kurt J. Lesker Company Ltd. (Hastings, UK), and the metal targets for sputtering (Cr, Pt) from Testbourne Ltd, (Basingstoke, UK) (all were of 99.99–99.999% purity). After metal deposition, ca. 6-μm-thick layer of the AZ4562 photoresist (Merck KGaA, Darmstadt, Germany) was spincoated (4000 rpm, 30 s), patterned by photolithography (1140 mJ/cm^2^) and developed in AZ726MIF developer (Microresist Technology GmbH, Berlin, Germany) for 5–6 min (Figure 1F). 

For metal etching (Figure 1G), case-specific etchants were used: Pt and Au were etched in aqua regia (a mixture of 69% nitric acid and 37% hydrochloric acid, 1:3, v/v) at 70 °C and room temperature, respectively. Ag was etched in a mixture of deionized water, hydrogen peroxide and ammonium hydroxide 12:1.8:1 (v/v). In case of Al, the phosphoric acid-based commercial etchant (PWS 80-16-4 (65), Honeywell, Charlotte, NC, USA) proved to be incompatible with Ormocomp (resulting in adhesion loss of Ormocomp from glass substrate) and instead, the Al etching was carried out in the basic AZ351B developer (Microresist Technology GmbH, Berlin, Germany). In addition to consecutive photoresist development in AZ726MIF developer (the standard process) and Al etching in AZ351B developer, each ca. 5–6 min each, both processes could be conducted simultaneously in AZ351B developer at room temperature by extending the immersion time to 30 min. Finally, the adhesion layer was etched by using either 15% (v/v) hydrogen peroxide containing 0.8% (v/v) ammonium hydroxide (for Ti) or 4% (v/v) perchloric acid containing 17% (m/m) cerium ammonium nitrate (for Cr) (Figure 1H). The chemicals used in metal etching were purchased from Honeywell (69% nitric acid, 37% hydrochloric acid, 30% hydrogen peroxide, and 25% ammonium hydroxide, all of the semiconductor grade VLSI PURANAL) or from VWR (70% perchloric acid, NORMAPUR, and ammonium cerium nitrate, RECTAPUR).

After metal etching, an oxygen plasma treatment of 1 min by reactive ion etching (125 mTorr, 150 W, O_2_ 40 sccm, Plasmalab 80, Oxford Instruments, Bristol, UK) was applied to restore the Ormocomp surface chemistry prior to bonding (Figure 1I), followed by resist removal in AZ-100 Remover (Microresist Technology GmbH, Berlin, Germany) (Figure 1J). Last, the metallized microchannel was sealed by adhesive bonding of a third, 20 μm-thick layer of Ormocomp spincoated (4000 rpm, 30 s) and flood exposed on top of a 3M transparency film (Figure 1K). The bonding process was conducted on a hotplate at 70–95 °C for 3 Min, followed by gradual cooling to room temperature (Figure 1L). In case of Al, thanks to the mild etching conditions in AZ351B developer, the oxygen plasma restoration could be omitted and the bonding done at room temperature.

### 2.3. Contact Angle Goniometry

The impact of the different etchants used and the oxygen plasma treatment on the Ormocomp surface properties were assessed by contact angle goniometry (Theta, Biolin Scientific, Espoo, Finland). The advancing and receding contact angles of the differently treated Ormocomp surfaces were in all cases measured before and after the chemical/plasma exposure. The advancing contact angles were measured by starting from a water droplet of 1 µL and increasing the volume to 6 µL with a pump rate of 0.1 µL/s. The receding contact angles were measured by starting from a water droplet of 6 µL and decreasing the volume to 0 µL with a pump rate of 0.1 µL/s. The reported values are averages of two measurements.

### 2.4. Adhesion Tests

The adhesion of each metal to Ormocomp was tested with and without the adhesion layers by the standard pull-off test [25]. A transparent adhesive tape (double sided, removable Scotch tape, 3M, Maplewood, MN, USA) of the size 30 × 20 mm^2^ was manually applied on top of the metallized surface and peeled off by hand, pulled fast and rigorously after ca. 5 s (Figure 2A). The quality of the metal adhesion was judged based on visual appearance under an optical microscope. Similar Scotch tape test was also applied for qualifying the polymer bonding strength. An additional ultrasound sonication test in water bath (5 min, 50 Hz, intensity setting 8, WPB ultrasound bath, Stangl Inc., Eichenau, Germany) was also performed for all metallizations. After ultrasound bath, the samples were dried by nitrogen stream and adhesion of metal on Ormocomp surface was again investigated by optical microscope.

### 2.5. Characterization of the Electrical Conductivity

To assess the electrical conductivity of the thin-film metallizations, each metal was deposited onto a planar, flood-exposed Ormocomp wafer and the sheet resistance (R_S_) of the thin-film was measured by a 4-point probe (CPS probe station, Keithley 2000 multimeter, Cascade Microtech, Beaverton, OR, USA) from the edges and the center of the wafer. The bulk resistivities (ρ) were calculated by multiplying the sheet resistance (average from n = 3 measurements) with the thickness of the metal layer, and compared with theoretical bulk resistivities.

A test set of metal wires of varying dimensions were also fabricated on top of the Ormocomp microchannels from each metal to be able to evaluate the quality of the step coverage. The measured resistances of the wires were compared with theoretical values calculated with help of the bulk conductivities and the wire dimensions. 

### 2.6. Microchip Electrophoresis with On-chip Amperometric Detection 

To demonstrate the feasibility of the developed metallization process for implementation of electrical-sensing elements with Ormocomp microchannels, a fully enclosed electrophoresis separation chip with integrated metal (Pt) electrodes was fabricated. The electrophoresis chip comprised of a 35 mm-long separation channel (150 μm × 35 μm, width × height) intersected by a 10 mm-long injection channel. The thin-film Pt electrodes (in this case, 220 nm thick layer on top of 17 nm Cr) were patterned at the separation channels outlet and designed so as to ensure compatibility with a commercial HVStat bipotentiostat (MicruX Technologies, Oviedo, Spain). The effective separation length from the channel cross-section to the separation channel outlet was 30 mm. The working, reference and auxiliary electrodes were 50 µm, 250 µm, and 250 µm wide, respectively, and 100 µm apart from each other (Figure 2B). 

Before analysis the microchip electrophoresis channel was sequentially rinsed with deionized Milli-Q water (Millipore, Bedford, UK) and the separation buffer (20 mM 2-(*N*-Morpholino)ethanesulfonic acid, MES, pH = 6.5) for 10–15 min each. The separation buffer was prepared in deionized Milli-Q water and the pH was adjusted with concentrated (2 M) sodium hydroxide. The stock solution of the sample (acetaminophen) was prepared in ethanol and diluted with the separation buffer to a final concentration of 20 μM. The MES hydrate, acetaminophen and ethanol were from Sigma-Aldrich (Steinhem, Germany), and sodium hydroxide from Riedel-de-Haen (Seelze, Germany).

The sample loading was performed by applying the sample to the sample inlet and the electric field (300 V/cm) between the sample inlet and outlet for 10 s, after which the sample was injected for separation by switching the potential difference (290 V/cm) between the separation channel inlet and outlet (Figure 2B). The amperometric detection of acetaminophen was performed using working electrode potential of 0.8 V. 

## 3. Results

### 3.1. General Notes on Ormocomp Metallization Possibilities

In this study, case-specific metallization processes were developed for Pt, Au, Ag, and Al with a view to integration of metal sensing elements with Ormocomp-based microfluidic devices. In particular, the challenges associated with the stability of the metallization (adhesion), step coverage (over microchannel edges), and polymer bonding over the metallized surface were addressed and successfully resolved. 

In terms of metal patterning, the focus was put on development of etching processes, since the alternative lift-off processes proved to be prone to adhesion loss. For example, immersion of the metallized Ormocomp surfaces in Mr-rem 400 or acetone solutions typically used for the photoresist (AZ4562, 6 µm) lift-off resulted in substantial loss of the metal adhesion. Although somewhat better adhesion was obtained by using AZ-100 remover as the lift-off solvent, the process was time-consuming (24 h at room temperature) and left rough edges and patches of residual metal that were difficult to remove. Instead, the etching-based metallization processes developed in this study survived incubation in an ultrasound bath of water without any visible damage, even if the adhesion layer was not deposited under the metal layer. The Scotch tape tests, however, revealed clear improvement in metal adhesion upon use of the adhesion layer. Only Ag showed equally strong adhesion to Ormocomp with and without the adhesion layer. 

### 3.2. Step Coverage and Polymer Bonding over Metal

Owing to the residual layer formation at the bottom of the microchannel upon controlled overexposure of Ormocomp, the resulting cross-section profile features tilted sidewalls (Figure 3A), which facilitates deposition of uniform and continuous metal layers across the Ormocomp channels (Figure 3B). The heavily rounded profile also helps in achieving better step coverage during metal deposition. Compared with the negative cross-section profiles typically obtained with other negative photoresists, the tilted sidewalls are also beneficial for patterning the photoresist (mask) required for the metal etching process. Namely, vertical sidewalls are difficult to expose and residual photoresist often remains at the bottom corners if these are sharp or negatively sloped. Moreover, the thickness of the metal layer also plays a role in terms of the quality of the step coverage (over microchannel edges). The thicknesses of the metal layers exposed to the pull-off tests were 30 nm, which was considered ideal for subsequent sealing of the metallized structures (by bonding with an Ormocomp lid) owing to the relatively small step height between metallized and non-metallized areas. Thin films of this thickness easily suffice for, e.g., optical applications, where Al or Ag films could act as mirror surfaces. However, to ensure elimination of any discontinuity in electrical-sensing elements, thicker metal layers are often required for achieving proper step coverage. However, as the layer thickness increases, the longer the etching time, which may adversely impact the metal adhesion. From this perspective and with a view to their prospected use in electrochemical applications, a layer thickness of 220 nm was considered the best compromise for the Pt electrodes, taking into account both continuity of the metal layer and the quality of the Ormocomp bonding over metal. For Au films, however, a layer thickness of 30 nm was found sufficient to ensure proper electrical conductivity as well (for more information, see Section 3.3). 

In addition to the step height (of the metal layer), the metal etchants were shown to have a substantial impact on the Ormocomp surface properties and thus also the bonding quality. The impact of the etchants on the wetting properties was examined with the help of contact angle goniometry. Compared with the native Ormocomp (advancing contact angle 77° ± 12°, receding contact angle 57° ± 10°), the contact angles after all metal etching protocols were generally much lower, but also non-homogenous making accurate goniometric analysis impossible. It was also observed that such non-homogeneous surface was not amenable to bonding in any of the cases. However, with help of an additional plasma oxidation step, the surface properties could be standardized and proper bonding quality assured. Upon plasma treatment, both advancing and receding contact angles dropped to nearly 0°, indicating high and uniform surface energy, which favors the bonding process. Nevertheless, the bonding had to be done at an elevated temperature (here, between 70 and 95 °C) to ensure proper sealing. Only in the case of the Al process, could the bonding be done at room temperature and similar bonding quality was reached with and without the plasma oxidation, likely because of the milder etching conditions (in the AZ351B resist developer) compared with other metals. The bonding strength in all cases was examined in a similar way to metal adhesion by using of the Scotch tape test. 

The optimized parameters for metallization and bonding are summarized in Table 2. As long as these conditions were followed, proper bonding quality similar to Figure 3C could be achieved. There are, however, a few noteworthy material-specific aspects. First, it should be noted that the Al etching process (in the AZ351B resist developer) often resulted in a non-uniform etch profile unless the adhesion layer (Cr) was used, as illustrated in Figure 3D. The non-uniformity of the etch profile was unambiguously associated with the extended solvent exposure time, as similar non-uniformity was not observed for any other metal, regardless of the adhesion layer. Second, care should be taken not to remove the photoresist prior to the plasma oxidation step in order to avoid oxidation of Ag. For other metals, the order of plasma oxidation and resist removal was less critical. Last, it should also be noted that the optimal temperature of the aqua regia etch (Au, Pt) was lower than recommended (70 °C). Namely, proper sealing of metallized Ormocomp was not obtained if the etching was conducted at 70 °C, but only after lowering the temperature of the aqua regia bath to room temperature (Au) or 55 °C (Pt).

### 3.3. Electrical Conductivity 

The electrical conductivity of the metallizations was examined in two different ways, based on sheet resistances of planar thin-film metals and by fabricating a test set of wires of each metal over Ormocomp microchannels (similar to Figure 3B) to account for the quality of the step coverage. The wires crossed the channel either one (wire length 15 mm) or three times (wire length 75 mm, meandering). Three different line (wire) widths, i.e., 50, 100 and 150 μm, were also tested. Although Ag and Al metallizations were primarily developed with a view to optical (micromirror) applications, they were also characterized with respect to the electrical conductivity along with Pt and Au. However, in this context, somewhat thicker metal layers of Ag (100 nm) and Al (140 nm) were deposited to ensure proper step coverage. The optimal layer thicknesses of Pt and Au were 220 nm and 30 nm, respectively, as discussed previously. The measured sheet resistances of the planar metal films were 8.9 ± 0.5 Ω (Pt 220 nm), 1.7 ± 0.1 Ω (Au 30 nm), 0.5 ± 0.1 Ω (Ag 100 nm), and 6.0 ± 0.5 Ω (Al 140 nm). The resistivities of Au and Ag films, derived from the sheet resistance values, were somewhat greater than the bulk resistivities [26], differing by a factor of two or three, respectively, which is typical for many metal thin films. However, the resistivities of Pt and Al films were about an order of magnitude greater (by a factor of 20 and 30, respectively) than the bulk resistivities, indicating that the Pt and Al deposition processes might lead to alloying with the adhesion layer or partial intermixing of the metal with the Ormocomp surface. However, all metal films, including Pt and Al, were clearly conductive, which was also evidenced by the measured wire resistances that were all in the range of 10^2^–10^3^ Ω (depending on the wire dimensions). Even when traveling across the edges of the microfluidic channels one or multiple times, the resistivities calculated based on the measured wire resistances were in good accordance with those calculated based on the measured sheet resistances of planar metal films on Ormocomp surfaces (Figure 4). These results clearly evidence that the step coverage of the metallizations, which was the primary goal in this study, was good and the conductivity of the metal wires was not noticeably affected by topographical features of the substrate surface.

### 3.4. Amperometric Detection of Acetaminophen

The feasibility of the developed metallization processes for implementation of metal sensing elements as integral parts of the Ormocomp-based microfluidic devices was eventually demonstrated with the help of amperometric detection of acetaminophen following on-chip electrophoresis. This combination of methods was chosen for the proof-of-concept experiments owing to its challenging nature. Namely, in electrochemical assays, the metal patterns are exposed to galvanic corrosion, and thus only noble metals (such as Ag, Au, or Pt) can be used as the electrode material. However, since Pt wires are typically used for application of the electrophoresis voltages (at the fluidic inlets and outlets), even Ag and Au metal films may suffer from galvanic corrosion, depending on the composition of the background electrolyte solution. This was also confirmed by our own experiments (data not shown). Therefore, Pt was the material of choice for the amperometric electrodes patterned at the outlet of the separation channel (Figure 5A). Still, the main metal layer has to be uniform to ensure that the underlying adhesion metal (Cr) will not etch away upon application of the electric field. In this work, no damage due to galvanic corrosion was observed on the Pt electrodes, when the optimized metallization protocol was followed (Figure 5B). The resulting electropherogram of acetaminophen run through the electrophoresis channel and detected by on-chip amperometry is presented in Figure 5C. The average (n = 5 repeated runs) of the migration time and peak area were 29.8 ± 0.7 s (2% relative standard deviation (RSD)) and 15.6 ± 2.8 a.u. (18% RSD). The repeatability of acetaminophen analysis (migration time 2% RSD, peak area 18% RSD) was similar to that obtained by using a commercial SU-8/glass hybrid electrophoresis chip of the same type (migration time 0.7%, peak area 10%, n = 6 runs of 10 μM acetaminophen). On the commercial chip (Pt001T, MicruX Technologies, Oviedo, Spain), the metal electrodes are patterned on top of a glass substrate plate and an SU-8 microfluidic channel is patterned on top, otherwise the microchannel and electrode designs were similar to this work.

## 4. Discussion

In this study, the Ormocomp metallization possibilities were comprehensively examined and new metallization processes were developed for Pt, Au, Ag, and Al. Through careful optimization of the etching conditions, strong adhesion and good step coverage over microchannel edges as deep as 35 μm was obtained for each of the metals. It was also concluded that the bonding and metallization parameters are interdependent. Although fabrication of enclosed Ormocomp microdevices has been reported in previous literature, the process was shown to be sensitive to alterations in the physico-chemical surface properties upon metal etching and needed to be customized for each metal separately. Uniform step coverage over microchannel edges poses another challenge, which was not thoroughly addressed until the present work. In this respect, Ormocomp provides unique opportunities for fabrication of tilted sidewalls in a single lithographic step, which facilitate deposition of uniform metal layers across the microchannel and help to avoid discontinuity often associated with three-dimensional metallizations. 

Overall, the developed processes enable implementation of a range of sensing elements without sacrificing the quality of the polymer bonding (sealing). Owing to the high reflectivity of Al and Ag, the prospected future use of these processes appears through implementation of non-planar mirror elements as integral parts of microfluidic Ormocomp devices. The Au and Pt metallization processes, in turn, are likely to facilitate integration of electrical (e.g., impedance) or electrochemical (e.g., amperometric) detector elements with Ormocomp-based cell culturing and chemical analysis devices, respectively. In this study, the proof-of-concept experiments were conducted with a view to electrochemical sensing of small molecules by demonstrating the fabrication and performance testing of an enclosed electrophoresis separation chip incorporating three integrated electrodes feasible for on-chip amperometric detection (of acetaminophen). The results indicated that equally good performance compared with commercial microdevices of the same type can be achieved. 

## Figures and Tables

**Figure 1 micromachines-10-00605-f001:**
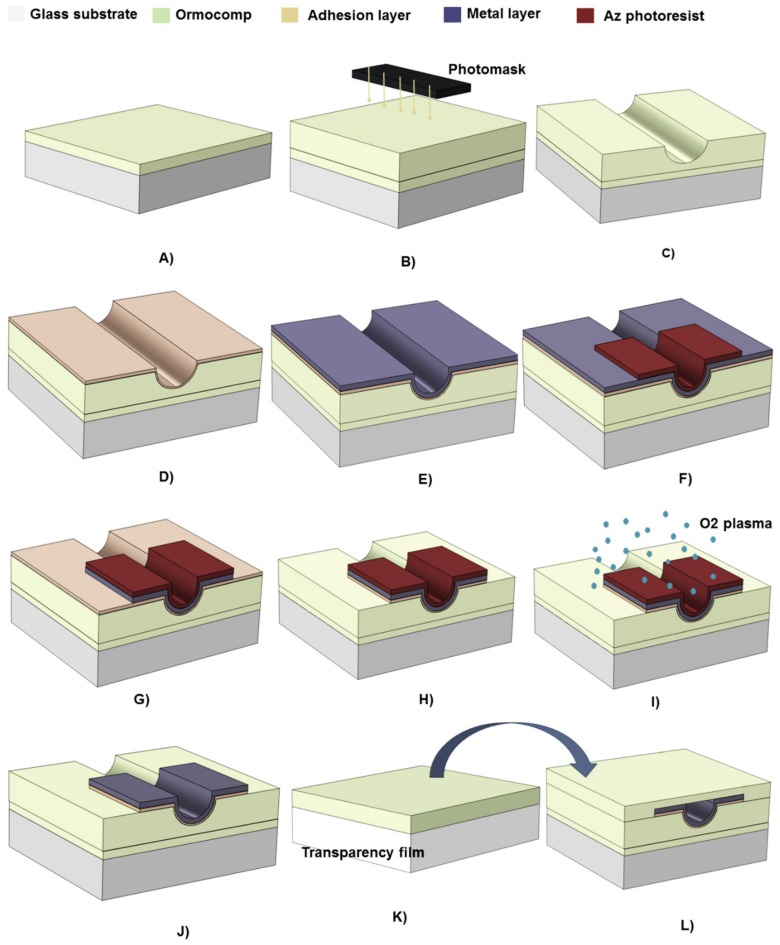
Schematic presentation of the photolithographic fabrication and metallization of concave Ormocomp microchannels: (**A**) spincoating and flood exposure of the first (bottom) layer of Ormocomp, (**B**) spincoating and masked exposure of the second (microchannel) layer, (**C**) Ormocomp development and hard bake, (**D**) deposition of the adhesion layer (17 nm Cr or 5 nm Ti), (**e**) deposition of the metal layer (Ag, Au, Al, or Pt) and hexamethyldisilazane coating, (**F**) spincoating and patterning of the photoresist, (**G**) etching of the metal layer, (**H**) etching of the adhesion layer, (**I**) oxygen plasma treatment, (**J**) photoresist removal, (**K**) spincoating and flood exposure of the third (bonding) layer of Ormocomp on a transparency film, and (**L**) adhesive bonding over the metallized microchannel.

**Figure 2 micromachines-10-00605-f002:**
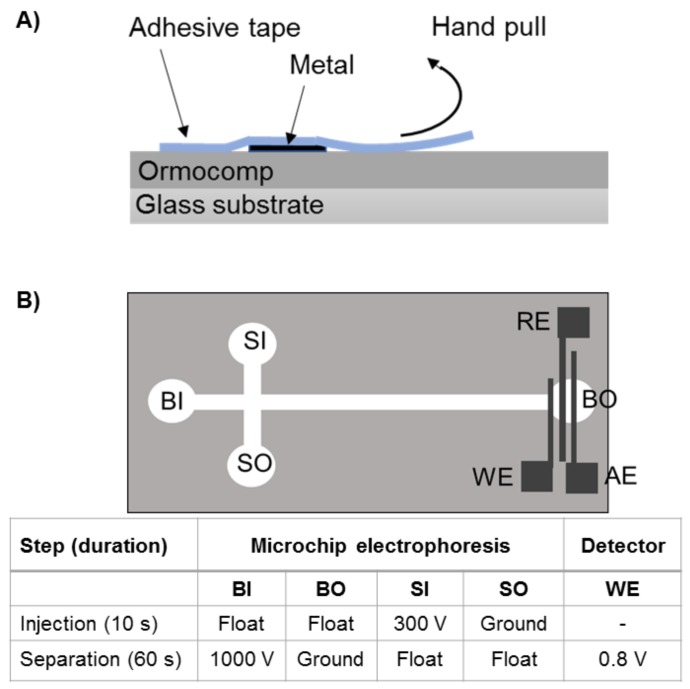
Schematic views of (**A**) the Scotch tape based adhesion test adopted from a previous work [25] and (**B**) the Ormocomp electrophoresis separation chip featuring Pt electrodes for on-chip amperometric detection, accompanied by a tabular presentation of the injection, separation, and detector voltages used in the proof-of-concept experiments. BI = buffer inlet, BO = buffer outlet, SI = sample inlet, SO = sample outlet, WE = working electrode, RE = reference electrode, AE = auxiliary electrode. In the electrophoresis experiments, the BO also served as the auxiliary electrode for the amperometric detection.

**Figure 3 micromachines-10-00605-f003:**
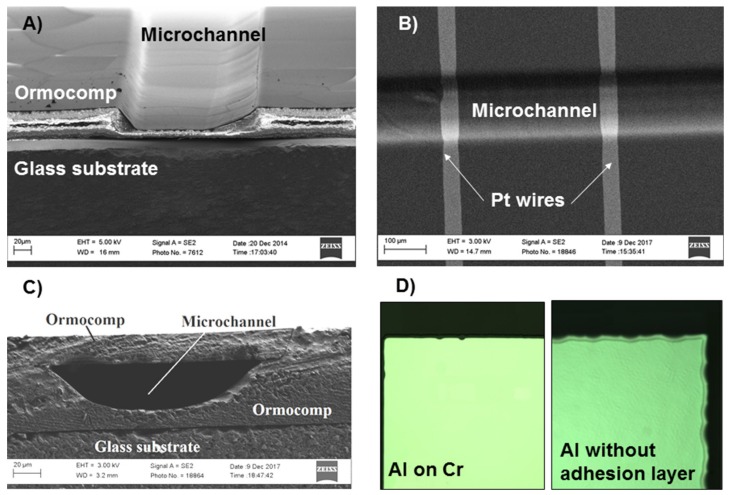
A scanning electron micrographs of (**A**) the cross-section of Ormocomp microchannel fabricated exploiting controlled overexposure (≥19 mJ/cm²) and featuring tilted side walls, (**B**) Pt wires (50 µm wide, 220 nm thick) crossing an Ormocomp channel, and (**C**) the cross-section of a bonded Ormocomp microchannel. (**D**) Comparison of the etch profile of Al thin-film metal in AZ351B developer with and without adhesion layer (17 nm Cr).

**Figure 4 micromachines-10-00605-f004:**
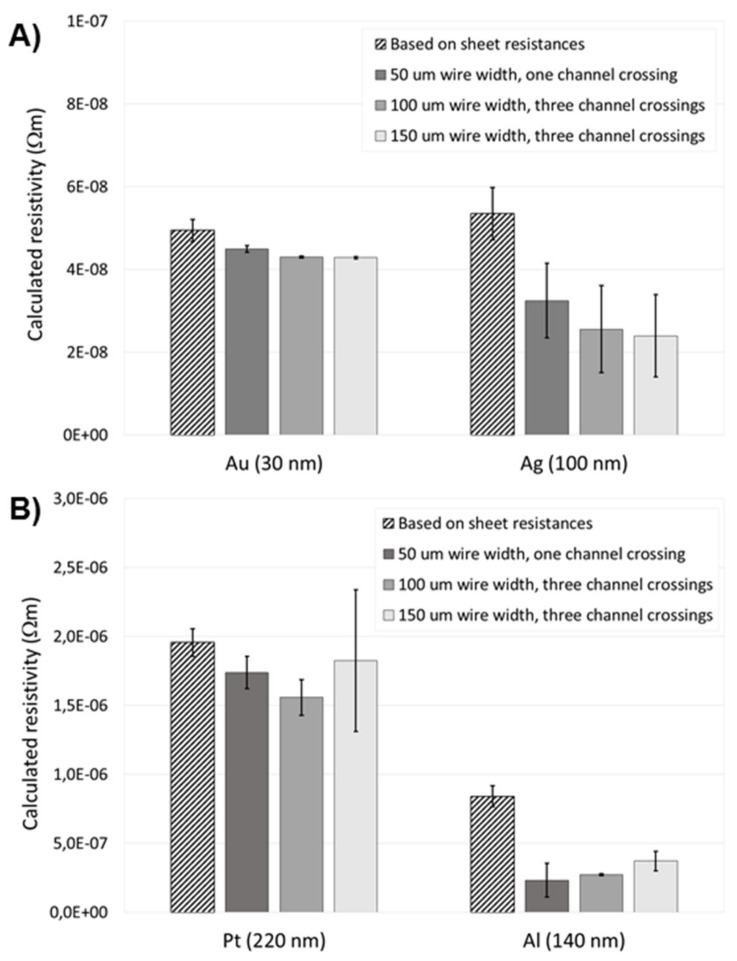
The resistivities of (**A**) Au and Ag metallizations and (**B**) Pt and Al metallizations calculated based on both measured sheet resistances of planar metal films and measured resistances of the metal wires (of different widths) travelling across the edges of Ormocomp microchannels one or three times.

**Figure 5 micromachines-10-00605-f005:**
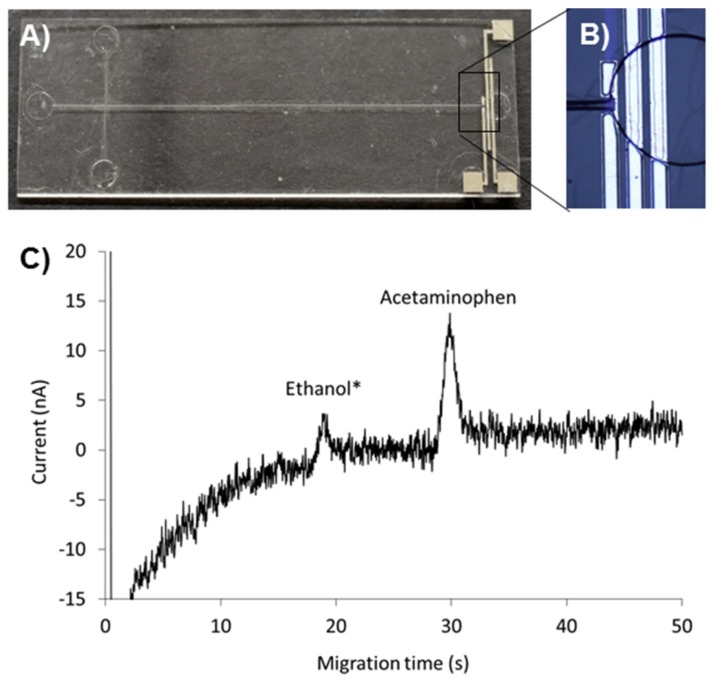
(**A**) Photograph of the Ormocomp electrophoresis separation chip featuring (**B**) Pt electrodes for on-chip amperometric detection. (**C**) Electrophreogram of 20 µM acetaminophen in 20 mM 2-(*N*-Morpholino)ethanesulfonic acid (MES) (pH = 6.5) buffer. The electric field strength during the electrophoresis separation was 290 V/cm and the working electrode potential 0.8 V. (*****) The ethanol peak is due to the residual solvent (0.2% ethanol) originating from the acetaminophen stock solution.

**Table 1 micromachines-10-00605-t001:** The parameters used for metal deposition by sputtering (Cr, Pt) and evaporation (Ti, Au, Ag, Al).

Metal	Evaporation Parameters		
	Current (mA)	Pressure (×10^−10^ bar)	Rate (Å/s)
**Ti**	60	5.0	0.5
**Au**	40	0.9	1.1
**Ag**	32	2.0	2.3
**Al**	15	1.0	1.2
**Metal**	**Sputtering Parameters**		
	**DC power (W)**	**Argon (sccm)**	**Time (s)**
**Cr**	200	50	105
**Pt**	500	70	480

**Table 2 micromachines-10-00605-t002:** The optimized metallization and bonding conditions.

Metal (nm)	Etching Conditions		Bonding Conditions	
	Etchant	Time	O_2_ Plasma	Temperature
Pt (220)	aqua regia (55 °C)	3–4 min	Yes	at 70–95 °C
Au (30)	aqua regia (RT)	10 s	Yes	at 70–95 °C
Ag (30)	0.8% NH_4_OH + 15% H_2_O_2_ in H_2_O	10 s	Yes (before resist removal)	at 70–95 °C
Al (30)	AZ351B developer	30 min	No	at RT
Ti (5)	0.8% NH_4_OH + 15% H_2_O_2_ in H_2_O	1–2 min	adhesion metals	
Cr (17)	17% Ce(NH_4_)_2_(NO_3_)_6_ + 4% HClO_4_ in H_2_O	7–10 s		

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
