# Peer review of "Metallization of Organically Modified Ceramics for Microfluidic Electrochemical Assays"

_micromachines, 2019, doi:10.3390/mi10090605_

Round 1
Reviewer 1 Report
The manuscript by Bonabi et al. is well-written and clearly presented. The text is easy to follow and the recipe for a functional metallisation process will be useful for the scientific community. A quick literature search reveal that other methods for metal coating of ormocomp exist, although they do seem more complicated. Nevertheless, it would be very useful if the author would include a short list of other methods from the literature, either in the introduction or as a separate table.
Author Response
The other methods reported for metal coating of Ormocomp are indeed very limited, but these are now listed in the Introduction. In addition to those papers originally listed in the manuscript (references 9, 13 and 14), we have added two more (new) references. A brief discussion of these papers is also added about these papers (page 2, lines 56-58 and 65-66) in line with the reviewer's request.
All changes made with respect to the original text are indicated with track-change.
Reviewer 2 Report
The article "Metallization of organically modified ceramics for microfluidic electrochemical assays" presents in-depth study of the process of metallization of the organically modified ceramics . The article is first of its kind to deal with ORMOCERs metalization process in great details, and as such will be of interests to the wider audience of Micromachines journal. The most valuable section of the article is materials and methods. The article have a great pedagogical value, as it teaches users how to perform metalization of ORMOCERs and it has applications beyond microfluidics. As such, this reviewer recommends publication of the article with minor revisions as specified below and believes that the article will become a highly cited neo-classical reference in the field of ORMOCERs processing. In order for process to be reproducible, before publications, the authors needs to expand with a more details next sections of materials and methods. The reviewer considers addition of this items minor revision and does not require secondary review:
1. In a section 2.2. Metallization and Bonding, each step needs to be described in a much greater details. The authors needs to specify exact equipment used, materials and its suppliers and and exact conditions for sputtering and evaporations. PLease describe the spin coating setup and conditions. Please name suppliers of all reagents used, as well as reagents grades. This is really the key section of the article. Needs to be expanded, with all details for full replication.
2. In section 2.3. Contact angle measurements please ellaborate on exact experimental conditions.
3. For the section 2.4. adhesion test, please describe in details adhesion test setup. It will be benefical to have a little illustration/drawing of principles of adhesion test. The exact size, and brand of Scotch tape used; The pulling force, or if just hand pull, describe it so. How exactly you quantify or semi-quantify adhesion results ? Please include some references for used method. Please describe in details ultrasonication testing of adhesion. How it is done? What is the medium ? For how long exposure ? What is the full power and frequency (based on model used). Specify complete model and manufactrer (name, location) of the ultrasonics bath used. How did you quantify surface ? How did you dry sample after the bath ?
After answering above, the article should be published.
Author Response
We have revised the manuscript according to the reviewer's requests (points 1-3 below):
1. Section 2.2. Metallization and Bonding:
Each step is now described in a greater detail by specifying the exact equipments and materials (incl. supplier names and grades) used in order to facilitate full replication. In addition, the exact sputtering and evaporation conditions are now given in a tabular form (Table 1) on page 4.
2. Section 2.3. Contact angle goniometry:
The exact experimental conditions and now better elaborated to meet the standard depth of details for this specific technology.
3. Section 2.4. Adhesion test:
As requested by the reviewer, the details of the adhesion test setups are now elaborated in the text for both tests (Scotch tape and ultrasound sonication), including the principle of (semi)quantification of the test result. In addition, literature reference and an illustration of the test principle (new Figure 2a) have been added for the Scotch tape test.
All changes made with respect to the original text are indicated with track-change.